# High Galectin-7 and Low Galectin-8 Expression and the Combination of both are Negative Prognosticators for Breast Cancer Patients

**DOI:** 10.3390/cancers12040953

**Published:** 2020-04-12

**Authors:** Anna Trebo, Nina Ditsch, Christina Kuhn, Helene Hildegard Heidegger, Christine Zeder-Goess, Thomas Kolben, Bastian Czogalla, Elisa Schmoeckel, Sven Mahner, Udo Jeschke, Anna Hester

**Affiliations:** 1Department of Obstetrics and Gynecology, University Hospital, Ludwig-Maximilans-Universität (LMU) Munich, 81377 Munich, Germany; Anna.trebo@campus.lmu.de (A.T.); nina.ditsch@uk-augsburg.de (N.D.); christina.kuhn@med.uni-muenchen.de (C.K.); helene.heidegger@med.uni-muenchen.de (H.H.H.); christine.goess@med.uni-muenchen.de (C.Z.-G.); thomas.kolben@med.uni-muenchen.de (T.K.); bastian.czogalla@med.uni-muenchen.de (B.C.); sven.mahner@med.uni-muenchen.de (S.M.); anna.hester@med.uni-muenchen.de (A.H.); 2Department of Obstetrics and Gynecology, University Hospital Augsburg, 86156 Augsburg, Germany; 3Institute of Pathology, LMU Munich, 81377 Munich, Germany; elisa.schmoeckel@med.uni-muenchen.de

**Keywords:** breast cancer, galectins, galectin-7, galectin-8, prognostic markers, tumor infiltrating macrophages

## Abstract

Galectins are commonly overexpressed in cancer cells and their expression pattern is often associated with the aggressiveness and metastatic phenotype of the tumor. This study investigates the prognostic influence of the expression of galectin-7 (Gal-7) and galectin-8 (Gal-8) in tumor cell cytoplasm, nucleus and on surrounding immune cells. Primary breast cancer tissue of 235 patients was analyzed for the expression of Gal-7 and Gal-8 and correlated with clinical and pathological data and the outcome. To identify immune cell subpopulations, immunofluorescence double staining was performed. Significant correlations of Gal-7 expression in the cytoplasm with HER2-status, PR status, patient age and grading, and of Gal-8 expression in the cytoplasm with HER2-status and patient age and of both galectins between each other were found. A high Gal-7 expression in the cytoplasm was a significant independent prognosticator for an impaired progression free survival (PFS) (*p* = 0.017) and distant disease-free survival (DDFS) (*p* = 0.030). Gal-7 was also expressed by tumor-infiltrating macrophages. High Gal-8 expression in the cytoplasm was associated with a significantly improved overall survival (OS) (*p* = 0.032). Clinical outcome in patients showing both high Gal-7 and with low Gal-8 expression was very poor. Further understanding of the role of galectins in the regulation and interaction of tumor cells and macrophages is essential for finding new therapeutic targets.

## 1. Introduction

Breast cancer is one of the three most prevalent cancers worldwide and the most frequent malignant tumor in women [1]. In 2018, about 2.1 million female patients were diagnosed with breast cancer and it represented the leading cause of tumor-related deaths in over 100 countries [2].

Based on gene profiling studies, breast cancer can be classified into different intrinsic subtypes [3]. In daily clinical practice, a surrogate system based on immunohistochemical and molecular characteristics is generally used: luminal A-like tumors show a strong expression of estrogen (ER) and progesterone (PR) receptors, low proliferation rates and a good prognosis. Luminal B-like human epidermal growth factor receptor 2 (HER2), and negative tumors show a lower hormone receptor expression and higher grading than Luminal A-like tumors and have an intermediate prognosis. Tumors that show an amplification of HER2 can be further classified based on hormone receptor expression in Luminal B-like HER2+ and in HER2+ non-luminal tumors. Triple negative breast cancers (TNBC) lack the expression of both hormone receptors and HER2, and show the worst prognosis of all breast cancer subtypes [3,4]. Even though therapy has improved in recent years, [1] new therapeutic strategies aiming at specific targets are needed [5,6,7]. This is especially important for TNBC, because patients still have a poor outcome and cannot be treated with endocrine therapy or targeted therapies like anti-HER2-therapy [8].

The discovery of galectins in 1994 [9,10] led to investigations on their impact on tumor development, progression and metastasis [11]. Galectins are a group of proteins able to bind to β-galactoside-binding sugars, either by N-linked or O-linked glycosylation, and they share primary structural homology in their carbohydrate-recognition domains (CRDs) [10]. A total of 12 different human galectin coding genes were found, including two for galectin-9 [12]. Galectins are divided into the subgroups of dimeric galectins (galectin-1,-2,-7,-10,-13,-14) with two identical CRD subunits, tandem galectins (galectin-4,-8,-9,-12) with two distinct CRD subunits, and chimeric galectins (galectin-3) with one or even multiple subunits of the same type [10,12]. Galectins are commonly overexpressed in cancer cells and cancer-associated stromal cells [13]. This altered galectin expression often correlates with the aggressiveness of the tumor and the metastatic phenotype [14]. In breast cancer, a connection between diverse galectin expression patterns and different cancer characteristics—like a correlation of galectin-1 (Gal-1) expression with tumor grading—was found [11,15]. Recently, the role of Gal-1 and galectin-3 (Gal-3) was thoroughly investigated [14]. Silencing Gal-1 led to both impaired tumor growth and reduced metastasis in a breast cancer mouse model [16]. Furthermore, it was found that Gal-1 interacts with E-selectin and influences adhesion [17]. However, targeting Gal-1 still has not come to clinical practice because no strategy of a fully specific Gal-1 blocking has been established yet. Gal-3 was identified as a molecular signature of breast cancer [18] and also as a potential therapeutic target [19]. Galectin-9, however, was described as anti-tumorigenic with possible antimetastatic potential in breast cancer [20]. Galectin-7 (Gal-7), like Gal-1 and Gal-3, seems to have tumor-promoting effects: in a Gal-7 deficient mouse model, a delayed development of HER2+ breast cancer was observed. Further investigations showed a positive correlation of Gal-7 expression with the frequency of HER2+ breast cancer. Furthermore, an association of Gal-7 expression with increased lymph node axillary metastasis in HER2+ tumors was seen [21,22]. Another study revealed an augmented Gal-7 expression in aggressive molecular subtypes, notably in estrogen receptor negative breast cancer and in cell lines with a basal-like phenotype. High expression of Gal-7 caused a higher metastatic risk, rendering cancer cells more resistant to apoptosis in a mouse model. Gal-7 might be part of the p53-promoted cancer progression pathway [23]. It is not known if these effects are specific to Gal-7, as most of the studies focused on one galectin and did not compare all individual galectins with each other. There is no detailed analysis of all galectins and their specific effects in breast cancer and most galectins, like galectin-8 (Gal-8), have not been studied in detail. There are only limited data suggesting that silencing a Gal-8-dependent pathway might lead to impaired tumor growth, especially in TNBC [24].

In this study, the specific role of Gal-7 and Gal-8 in a bigger cohort of human primary non-metastatic breast cancer was evaluated. The analysis included the specific location of galectin expression in the nucleus or cytoplasm and the expression in tumor-surrounding immune cells.

## 2. Results

### 2.1. Gal-7 and Gal-8 Expression in Breast Cancer and Correlation to Different Clinical and Pathological Characteristics

#### 2.1.1. Gal-7 and Gal-8 Expression in Breast Cancer Cytoplasm and Nucleus

Expression of both galectins was observed in the cytoplasm as well as in the nucleus, being more pronounced for Gal-7. Gal-7 expression could not be evaluated in 19 sections (due to technical issues). Seven cases showed no Gal-7 expression in the cytoplasm and 35 no staining in the nucleus. For Gal-8, 20 tissue sections could not be analyzed (due to technical issues), and 15 and 63 samples revealed no staining in the cytoplasm and in the nucleus, respectively. The distribution of the immunoreactivity score (IRS) for the staining in the cytoplasm and in the nucleus is shown in Table 1; Table 2. The mean IRS for staining in the cytoplasm was 4.88 for Gal-7 and 4.37 for Gal-8, while it was 2.51 for staining in the nucleus for Gal-7 and 2.56 for Gal-8.

#### 2.1.2. Gal-7 and Gal-8 Expression and Correlation with Clinical Characteristics, Histopathological Breast Cancer Subtypes and Grading

Both Gal-7 and Gal-8 expression in the cytoplasm and in the nucleus did not correlate with the clinical parameters: tumor size and lymph node status. Gal-7 and Gal-8 expression in the cytoplasm correlated negatively with patient age (see Spearman correlation analyses in Appendix A).

The Gal-7 expression correlated negatively with the histopathological subtype. Kruskal–Wallis test and boxplots analysis showed that Gal-7 expression in the cytoplasm was significantly higher in no special type (NST) tumors compared to non-NST tumors (median IRS 6 in NST vs. 4 in non-NST, *p* < 0.001) (Figure 1a). Similarly, Gal-7 expression in the nucleus was significantly higher in NST tumors (*p* = 0.042). The Gal-8 expression did not differ concerning the histological subtype.

Regarding tumor grading, a positive correlation with the cytoplasmic Gal-7 expression was found (Spearman correlation analysis in Appendix A). Kruskal–Wallis test showed that Gal-7 expression in the cytoplasm was higher in higher tumor grading (Gal-7 in G1 median IRS 3 and in G2/3 median IRS 6, *p* = 0.003, Figure 1b). The Gal-7 expression in the nucleus and the Gal-8 expression in the cytoplasm were not associated with tumor grading. The Gal-8 expression in the nucleus correlated negatively with the tumor grading and showed a trend towards a lower IRS in higher tumor grading (*p* = 0.089). Exemplary immunohistochemical Gal-7 and Gal-8 staining in tumors with different gradings are shown in Figure 2. 

#### 2.1.3. Gal-7 and Gal-8 Expression and Correlation with Hormone Receptor Status, HER2 Amplification and Surrogate Intrinsic Subtypes

Spearman analysis revealed that Gal-7 expression in the cytoplasm did correlate to PR-status and Gal-8 expression in the nucleus to ER-status (Spearman analysis in Appendix A). In the Kruskal–Wallis analysis, the Gal-7 staining in the cytoplasm was significantly higher in PR-negative compared to PR-positive tumors (median IRS in PR-positive: 4 vs. in PR-negative: 6, *p* = 0.038, Figure 3b), but was not significantly different concerning ER-status (*p* = 0.159, Figure 3a). The Gal-8 expression in the nucleus was significantly higher in ER-positive tumors (*p* = 0.026, Figure 3c) and a trend towards a higher Gal-8 expression in PR-positive compared to PR-negative tumors was observed (*p* = 0.098, Figure 3d). Both Gal-7 expression in the nucleus and Gal-8 in the cytoplasm did not correlate with the ER- or PR-status.

Both Gal-7 and Gal-8 staining in the cytoplasm correlated significantly with the HER2-status (Spearman analysis in Appendix A). HER2-positive breast cancer samples showed a distinctly higher Gal-7 expression in the cytoplasm compared to HER2-negative tumor sections (median IRS in HER2+: 8 vs. in HER2-: 4, *p* < 0.001, Figure 3e). Gal-8 expression in the cytoplasm in HER2-positive tissue sections was significantly higher than in HER2-negative samples (median IRS in HER2+: 6 vs. in HER2-: 3, *p* = 0.004, Figure 3f). Gal-7 and Gal-8 staining in the nucleus were not significantly associated with HER2-status (Appendix A). 

The Gal-7 expression in the cytoplasm differed significantly regarding the different surrogate intrinsic subtypes (*p* < 0.001) (Figure 4): HER2-positive tumors (both luminal B-like and non-luminal) clearly showed the highest Gal-7 expression in the cytoplasm compared to all other subtypes (Luminal A-like and B-like and TNBC). The distribution of Gal-8 staining, neither in the cytoplasm (Appendix A) nor in the nucleus, and Gal-7 expression in the nucleus in the different surrogate intrinsic subtypes showed no significant differences.

#### 2.1.4. Correlation of Gal-7 and Gal-8 Expression

Gal-7 expression in the nucleus and cytoplasm correlated to each other as well as Gal-8 expression in the nucleus and cytoplasm (Table 3). Gal-7 and Gal-8 expressions in the nucleus and cytoplasm also correlated with each other.

### 2.2. Correlation of Gal-7 and Gal-8 Expression with Survival in Breast Cancer Patients

Median overall survival (OS), progression-free survival (PFS) and distant disease-free survival (DDFS) in the whole cohort was not reached (NR). The prognostic relevance of Gal-7 and -8 was analyzed concerning PFS, DDFS and OS and tumors were categorized in “high” and “low” expressing tumors using ROC-curve analysis.

#### 2.2.1. High Gal-7 Expression in the Cytoplasm is a Negative Prognosticator for Survival in Breast Cancer Patients

High Gal-7 expression in the cytoplasm (IRS > 6) was associated with a worse outcome: tumors with high Gal-7 expression in the cytoplasm showed a significantly impaired PFS (*p* = 0.017, median PFS in Gal-7 high: 9.7 years, median PFS in Gal-7 low: NR) (Figure 5a) and DDFS (*p* = 0.030, median DDFS in both subgroups NR) (Figure 5b). Concerning OS, no significant difference was detected (*p* = 0.927, median OS in both subgroups NR) (Figure 5c).

Multivariate COX regression analysis revealed Gal-7 expression as an independent prognostic factor for PFS (Table 4) but not for DDFS (Appendix A). Concerning OS, where Gal-7 was not significant in the univariate analysis, COX regression analysis revealed ER-status, nodal status and age as independent prognosticators (Appendix A).

Subgroup analysis revealed that Gal-7 expression in the cytoplasm had prognostic relevance for an impaired PFS in the ER-negative (but not in ER-positive) (*p* = 0.036, median PFS in Gal-7 high 2.52 years, Appendix A), in the pT1 (but not in pT > 1) (*p* = 0.014) and in the pN0 (but not in pN > 0) (*p* = 0.043) subgroups. A trend towards an impaired PFS could be observed in PR positive (but not PR negative) (*p* = 0.098) and in HER2 positive (but not in HER2 negative) tumors (*p* = 0.084, Appendix A). 

Gal-7 expression in the nucleus did not show prognostic relevance for PFS, DDFS or for the OS (Appendix A). Subgroup analysis did also not show any new prognostic relevance of nuclear Gal-7 expression concerning the subgroups of PR-positive vs. -negative, HER2-positive vs. -negative tumors, grading, tumor size and lymph node status. However, in ER-negative tumors, a high nuclear Gal-7 expression showed an impaired OS while nuclear Gal-7 expression was not relevant in ER-positive tumors even if statistical significance was not reached (*p* = 0.082. Similarly, a worse outcome regarding PFS could be seen in lower grading (G1, *p* = 0.058) (but not in G2-3) when nuclear Gal-7 expression was high.

#### 2.2.2. High Gal-8 Expression in the Cytoplasm is a Positive Prognosticator for Overall Survival in Breast Cancer Patients

High Gal-8 expression in the cytoplasm (IRS > 5) was associated with an improved outcome: Tumors with a high Gal-8 expression in the cytoplasm showed a significantly improved OS compared to tumors with low Gal-8 expression in the cytoplasm (*p* = 0.032, median OS in both subgroups NR) (Figure 6a). No significance in the outcome regarding PFS (*p* = 0.974, median PFS in both subgroups NS, Figure 6b) and DDFS (*p* = 0.138, median DDFS in both subgroups NR, Figure 6c) was found.

The multivariate COX regression analysis could not confirm Gal-8 expression in the cytoplasm as an independent prognostic factor (Table 5) for OS. In contrast, age at the surgery, ER status and lymph-node status (pN) were detected as independent prognosticators for the OS. Gal-8 was also not prognostically relevant in COX regression for PFS or DDFS.

Subgroup analysis revealed that Gal-8 expression in the cytoplasm had prognostic relevance with a better outcome for the OS in NST (but not non-NST) tumors (*p* = 0.049), in HER2-negative (but not HER2-positive) (*p* = 0.029, Appendix A) in ER-positive (but not ER-negative) (*p* = 0.055, Appendix A) and in pT1 (but not in pT2-4) (*p* = 0.038). Concerning PFS and DDFS, no significant prognostic impact could be found for Gal-8 in the different subgroups.

Similar to Gal-7 staining in the nucleus, Gal-8 staining in the nucleus did not show prognostic relevance for neither PFS, DDFS or OS (Appendix A). However, subgroup analysis revealed that high Gal-8 staining in the nucleus was associated with improved PFS in ER-positive patients (*p* = 0.041) but not in ER negative patients and to tendentially (*p* = 0.081) better OS in pT2-4 (but not in pT1) tumors. Regarding the other subgroups (PR-positive vs. -negative, HER2-positive vs. -negative, nodal status, histological subtype), nuclear Gal-8 staining did not show significant prognostic influence, similar to the overall cohort. 

#### 2.2.3. Survival Analysis Using Combined Gal-7 and Gal-8 Staining

Survival analysis was also performed in subgroups with high or low Gal-7 expression in the cytoplasm in combination with low or high Gal-8 expression in the cytoplasm, respectively. A high Gal-7 combined with a low Gal-8 expression in the cytoplasm was associated with an impaired OS compared to all other patients (*p* = 0.201, median OS in all subgroups NR, Figure 7a). When Gal-7 expression was high and Gal-8 was low, DDFS was significantly impaired (*p* = 0.009, median DDFS in all subgroups NR, Figure 7b) compared to the rest of the patients, as well as PFS with a borderline significance (*p* = 0.067, median PFS in all subgroups NR, Figure 7c). In summary, two different groups regarding the outcome could be defined: The first group, representing 6.8% of the patients, consisted of tumors expressing Gal-7 high (IRS > 6) and Gal-8 low (IRS ≤ 5) in the cytoplasm. This subgroup showed a poor OS. The second group, formed by 13.6% of the patients with high Gal-8 (IRS > 5) and low Gal-7 (IRS ≤ 6) expression in the cytoplasm, showed good OS (Figure 7a). Comparing the three subgroups concerning PFS and DDFS, the “advantageous” subgroup (high Gal-8, low Gal-7) seems to be less relevant (survival curves similar to the “others” subgroup, Appendix A).

#### 2.2.4. Immune Cell Infiltration Stained with Gal-7

Gal-7 had an interesting expression pattern in the tumor tissue, with staining on only the outer layer of cancer cells and expression in the immune cells next to the tumor cells as well (Figure 8a–d). Gal-7 staining in the immune cells was not included as part of the IRS-scoring for the cancer specimen in the analyses described above. The immune cell staining of Gal-7 correlated significantly with the tumor grading: in tissue sections defined as grade 1 no stained immune cell infiltration was observed, whereas in grade 2 and 3, stained immune cells were detected (*p* = 0.008, Appendix A). Gal-7-expressing immune cells were furthermore found significantly more often in NST compared to non-NST tumors (*p* = 0.001, Appendix A) and in tumors with lymph node metastasis (*p* = 0.038, Appendix Ac). A double immunofluorescence staining with Gal-7 and CD68 (a macrophage marker) showed that the Gal-7 stained immune cells were macrophages (Figure 8e–g).

## 3. Discussion

Galectins have been shown to be a pivotal factor in carcinogenesis. However, data on their relevance in breast cancer are still sparse. As recently proposed [22,25], we could show in our cohort of primary breast cancer patients that Gal-7 was a negative prognosticator for the clinical outcome. Gal-7 expression in the cytoplasm had a negative prognostic impact on PFS and DDFS, a negative trend on OS and was even an independent negative prognosticator for PFS in the multivariate analysis. A strong association between a high Gal-7 expression in the cytoplasm and HER2 amplification was observed, suggesting that HER2-positive tumors might be especially interesting for potentially targeting Gal-7. In HER2-positive tumors—like in the overall cohort—high Gal-7 expression was associated with an impaired prognosis. Gal-8 expression in the cytoplasm, however, was associated with an improved OS.

Our data are in line with Grosset et al., [21,25] who found high Gal-7 expression only in HER2-positive tumors and in TNBC. Recent studies showed Gal-3 as a modifier of the epidermal growth factor receptor (EGFR), which is a regulator of cell growth and survival in normal and cancerous tissues [26,27,28]. EGFR and HER2 are both known as members of the ErbB receptor family and, when activated, they stimulate the activation of many signaling pathways [29]. Apart from Gal-3, no functional associations between galectins and ErbB receptors have been reported to date. However, these data suggest a functional connection between these two families which might also exist between Gal-7 and HER2. 

Other groups showed in an in silico mRNA survival analysis that Gal-7 and Gal-8 did not have a prognostic impact in breast cancer patients. However, regarding protein level—similarly to our data—high Gal-7 expression was associated with an impaired PFS (although not statistically significant), whereas no effect could be shown for Gal-8 (both in the overall cohort) [25]. 

Our results suggest that combining Gal-7 and Gal-8 expression might further improve prognostic accuracy. Two different groups could be defined: a small group (high Gal-8, low Gal-7) with a very good outcome (6.8% of the patients), compared to a second group (high Gal-7, low Gal-8) with a worse outcome, consisting of 13.6% of the patients. This is especially interesting, as Gal-7 and Gal-8 expressions are strongly correlated. Similar effects have been shown for galectin ligands: in BC patients, high levels of- Gal-1 ligands and low levels of Gal-8 ligands have been observed, making their ratio a strong marker for BC [30]. A similar ratio could be established using the Gal-7/Gal-8 expression for prognostic considerations. 

Galectin expression was also associated with tumor cell differentiation: Gal-7 expression was significantly higher in less differentiated cells (reflected by higher grading), whereas Gal-8 expression was highest in highly differentiated cells (reflected by lower grading). This is similar to other data where high Gal-4 (which belongs to the same family as Gal-8) expression in highly differentiated, and low Gal-1 (which belongs to the same family as Gal-7) expression in poorly differentiated, tumors have been shown [31]. Furthermore, in our study, poor tumor cell differentiation was also associated with high Gal-7 expression in tumor-surrounding macrophages, who are known to correlate with poor breast cancer survival rates.

Regarding mechanisms of the regulation of galectin expression and distribution between nucleus and cytoplasm, it is important to consider that Gal-7 is also provided extracellularly: The extracellular Gal-7 controls the intracellular pool of Gal-7, firstly by an increase in the gene transcription, and secondly by a re-entry pathway into the cells [32]. Similarly, for Gal-1 (which belongs to the same group as Gal-7) an extracellular to nucleus transfer has been shown and nuclear Gal-1 accumulation drove epithelial invasiveness. Extracellular glycans that bear N-acetyllactosamins (LacNAc) epitopes bind Gal-1 and trap it extracellularly. An α-2,6-sialylation of these LacNAc epitopes inhibits Gal-1-binding and drives the nuclear transfer of Gal-1 [33]. We could observe in our study that Gal-7 was also present in macrophages next to the tumor cells. Therefore, these macrophages might also provide a source of extracellular Gal-7 for tumor cells and might regulate the intracellular Gal-7 pool. Tumor-associated macrophages are correlated with poor survival rates of breast cancer [34], rendering Gal-7 even more interesting as a therapeutic target. Functional effects that have been described for Gal-8 include the activation of the activated leukocyte cell adhesion molecule (ALCAM) [23,35,36] and the activation of the endothelial nitric oxide synthase (eNOS) pathway on endothelial cells [37]. However, these pathways all describe a tumorigenic potential of Gal-8, while Gal-8 was a positive prognosticator in our study. Therefore, other tumor-suppressing pathways that need further investigation might exist.

First attempts in targeting galectins in breast cancer have already been made: Grosset et al. demonstrated that targeting CRD-independent cytosolic Gal-7 in breast cancer cells, and therefore impairing p53-functions, might be a valuable strategy for the treatment of breast cancer [35]. Targeting Gal-8 has not been performed to date. Gal-3 knockdown enhanced the sensitivity of tumor cells to the apoptotic agent arsenic trioxide (ATO, which is already approved by the US Food and Drug Administration for the treatment of acute myeloid leukemia) [19], making Gal-3 an interesting therapeutic target. An orally applied Gal-3 antagonist was already studied in a mouse model, leading to less lung adenocarcinoma growth [36]. Furthermore, a Gal-1 inhibitor that showed synergistic activity with the chemotherapeutic paclitaxel in BC has been discovered [37]. As Gal-7 belongs to the same family as Gal-1, similar therapeutic potential could exist for Gal-7. 

## 4. Materials and Methods

### 4.1. Patients

Formalin-fixed, paraffin-embedded (FFPE) primary breast cancer samples of 235 patients were examined in this study (Table 6).

We included in our study all patients that were diagnosed with primary non-metastatic breast cancer and underwent surgery at the Department of Gynaecology and Obstetrics, Ludwig-Maximilians-University Munich, Germany, from the period 1998 to 2000. 

Therefore, there was no pre-selection and a complete group of patients attending the clinic was analysed. Only women with benign tumors of the breast were excluded from the study.

Clinical, pathological and follow-up data (up to ten years) were retrieved from patients’ charts and from the Munich Cancer Registry. Patient characteristics are displayed in Table 6. In terms of clinical and pathological characteristics and tumor biology, the collective represents the reality of the wide BC spectrum.

Histopathological subtype (no specific type (NST) vs. non-NST), tumor grading (G1-3) according to the Elston and Ellis criteria (1993) [4,38,39], and staging using the TNM-System [40] (T for tumor size, N for the lymph node status and M for metastasis) were determined by a gynecological pathologist. As tumor grading could only be obtained in about 70% of all patients, the results have to be regarded with limited reliability. HER2-positivity is defined by the DAKO Scoring system (DAKO, HER2 FISH pharmDx™ Kit, Agilent Technologies, Waldbronn, Germany). As HER2 status was not determined routinely in Germany before 2001, it was retrospectively assessed. HER2 status was determined as recommended in the national guidelines, i.e., by DAKO Score and FISH analysis in cases of DAKO 2+.

Endpoints regarding the survival data were defined as follows: OS = overall survival, period of time from the date of surgery until the date of death or date of last follow-up; PFS = progression free survival, period of time until local recurrence or metastasis were diagnosed and DDFS = distant disease free survival: period of time until metastasis was diagnosed.

### 4.2. Immunohistochemistry

Paraffine-embedded breast cancer tissue samples were analyzed by immunohistochemistry. The samples were fixed in neutral buffered formalin and embedded in paraffin after surgery. For histopathological investigations, tissue sections (3 µm) were deparaffinized in Roticlear (Carl Roth GmbH + Co. KG) for 20 min and then the endogenous peroxidase was inactivated with 3% hydrogen peroxide (VWR International GmbH) in methanol. The slides were rehydrated in a descending gradient of ethanol (100%, 75% and 50%) and prepared for epitope retrieval in a pressure cooker for 5 min in sodium citrate buffer (0.1 mol/L citric acid, 0.1 mol/L sodium citrate, pH 6.0). After washing in distilled water and phosphate-buffered saline (PBS), all tissue slides were blocked using a blocking solution (Reagent 1; ZytoChem Plus HRP Polymer System (Mouse/Rabbit); Zytomed Systems GmbH, Berlin, Germany) for 5 min at room temperature (RT) in order to block non-specific binding of the primary antibodies. Then, a specific procedure followed for each Galectin: the slides were incubated with Gal-7 primary antibody (rabbit, polyclonal; Abcam, ab10482) at a final concentration of 2.5 µg/mL in PBS Dulbecco (Biochrom GmbH) for 16 h at 4 °C. Gal-8 primary antibody (rabbit, monoclonal, Abcam, ab109519) was used for incubation at a final concentration of 3.3 µg/mL in PBS for 1 h at RT. Afterwards, the staining specimens were incubated in post block reagent (Reagent 2) and HRP-polymer (Reagent 3) containing secondary antibodies (anti-mouse/-rabbit) and peroxidase according to the manufacturer’s protocol. These antibodies are part of the provided Reagent 3, exact concentrations are not specified by the manufacturer. (Reagent 2 and 3, ZytoChem Plus HRP Polymer System, Mouse/Rabbit). All slides were washed in PBS after every incubation step. The slides were then stained with 3,3′-diaminobenzidine chromogen (DAB; Dako, Glostrup, Denmark) for visualization and counterstained in Mayer acidic hematoxylin. After dehydrating in an ascending ethanol gradient and Roticlear they were cover slipped with Roti-Mount (Carl Roth GmbH + Co. KG). Appropriate tissue slides were used as positive controls (sigma tissue for Gal-7 and placenta for Gal-8). To obtain expression results, the semiquantitative immunoreactive score (IRS, Remmele and Stegner 1987 [41] was performed using a Leitz Diaplan microscope (Leitz, Wetzlar, Germany). The score was optically obtained by multiplying the predominant staining intensity (0: none; 1: low; 2: moderate; 3: strong) and the percentage of positively stained cancer cells (0 = 0%, 1= 1–10%, 2 = 11–50%, 3 = 51%–80%, and 4 = 81%–100% stained cells). The IRS was determined separately in the cytoplasm and the nucleus of the cancer cells. The staining of other cells, like immune cells, was not included in the IRS. Images were taken with a CCD color camera (JVC, Victor Company of Japan, Japan).

Furthermore, immune cells stained with Gal-7 were analyzed independently of the tumor IRS for Gal-7 and evaluated using following scoring system: 0 = no immune cells were stained, 1 = <50% and 2 = >50% of the immune cells were stained.

### 4.3. Immunofluorescence

Paraffine-embedded breast cancer tissue samples were also used for immunofluorescence-analyses. The samples were fixed in neutral buffered formalin and embedded in paraffin after surgery. After deparaffinization in Roticlear (Carl Roth GmbH + Co. KG) for 20 min, the slides were rehydrated in a descending gradient of ethanol (100%, 75% and 50%) and prepared for epitope retrieval in a pressure cooker for 5 min in sodium citrate buffer (0.1 mol/L citric acid, 0.1 mol/L sodium citrate, pH 6.0). After washing in distilled water and PBS, all tissue slides were blocked with Ultra V Block (Thermo scientific) for 15 min. The primary antibodies were diluted in Dako Antibody Diluent (Dako North America) incubated with the slides for 16 h at 4 degrees. Gal-7 (rabbit, polyclonal; Abcam, ab10482) was diluted at a final concentration of 2,5 µg/mL and CD68 (mouse, monoclonal, Sigma AldrichAMAb90874) at a concentration of 0.1 µg/mL. Next, the light in the room was dimmed and an incubation with the secondary antibodies for 30 min at RT followed. Secondary antibodies: Goat-Anti Rabbit IgG Cy3 (Dianova/Jackson, 111-165-144) diluted at a concentration of 3 µg/mL and Goat-Anti-Mouse-AlexaFluor488-IgG (Dianova/Jackson, 115-546-062) at a concentration of 15 µg/mL. After the slices were dried in the dark, they were cover slipped with mounting medium for fluorescence with DAPI (Vectashield H-1200). The samples were then analyzed using a Zeiss AxioPhot microscope with an Axiocam MRm within one day.

### 4.4. Statistical Analysis

Data analyses were performed with SPSS Statistics 25 (Armonk, NY: IBM Corp.). *p*-values lower than 0.05 were considered as statistically significant. Correlations between staining results and ordinal variables were tested with Spearman’s rank correlation coefficient. Group comparisons regarding the IRS of galectins between different clinical and pathological subgroups were tested with Kruskal–Wallis test and displayed as boxplot graphs. Survival times between different groups were compared by Kaplan–Meier analysis, and differences were tested for significance by Log-Rank (Mantel-Cox), Breslow- and Tarone-Ware-tests. Censored cases are cases for which the second event is not recorded (for example, people still alive at the end of the study). The Kaplan–Meier procedure is a method of estimating time-to-event models in the presence of censored cases. The Kaplan–Meier model is based on estimating conditional probabilities at each time point when an event occurs and taking the product limit of those probabilities to estimate the survival rate at each point in time. Cox-regression analysis was used to determine the independence of prognostic factors.

Concerning survival analysis dependent on Gal-7 and Gal-8 expression, patients were grouped into high and low expression. Cut-off points were selected considering the distribution pattern of IR-scores in the collective. Therefore, the receiver operator curve (ROC curve) was drawn using SPSS software, which is considered as one of the most reliable methods for cut-off point selection. In this context, the ROC curve is a plot representing sensitivity on the y-axis and (1-specificity) the x-axis. Consecutively Youden index, defined as the maximum (sensitivity + specificity−1), was used to find the optimal cut-off maximizing the sum of sensitivity and specificity (exemplary results of the ROC curve analysis in Appendix A). Furthermore, the medians of Kruskal–Wallis tests were observed and evaluated in order to find the ideal cutoff. The cytoplasmatic Gal-7 expression was regarded as low with an IRS 0–6 and as high with an IRS > 6. The cytoplasmatic Gal-8 expression was regarded as low with an IRS 0–5 and as high with an IRS > 5. 

### 4.5. Ethics Approval and Consent to Participate

This study has been approved by the Ethics Committee of the Ludwig-Maximilian-University Munich (approval number 048–08). The breast cancer specimens were obtained in clinically indicated surgeries. When the current study was performed, all diagnostic procedures were completed, and the patients’ data were anonymized. The ethical principles adopted in the Declaration of Helsinki 1975 have been respected. As per the declaration of our ethics committee, no written informed consent of the participants or permission to publish is needed given the circumstances described above. Researchers were blinded from patient data during experimental and statistical analysis.

## 5. Conclusions

In summary, our results suggest that Gal-7 might be an independent negative prognostic factor in breast cancer and therapeutic target, especially in HER2-positive breast cancer. Furthermore, Gal-8 was observed to be a positive predictor for overall survival and upregulation should be further investigated. The role of Gal-7 and Gal-8 should be validated in a BC collective treated with today’s standard of therapy—even if this might be outdated at the time point of the analysis. Additional studies are required to detect the signaling pathways in which both Gal-7 and Gal-8 are involved, as the combination of both markers showed strong prognostic impact. Gal-7 and Gal-8, as well as the whole group of galectins, seem to be interesting therapeutic and prognostic targets that might help to improve therapies and outcome for breast cancer patients in the future.

## Figures and Tables

**Figure 1 cancers-12-00953-f001:**
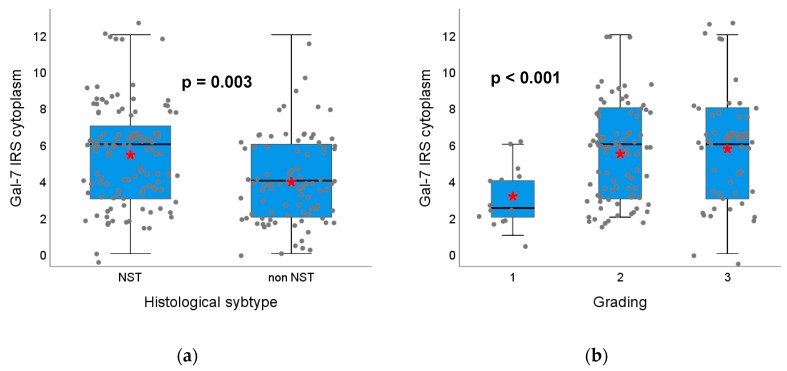
Association of Gal-7 expression with histological subtype and tumor grading. Boxplots of the median IRS of Gal-7 staining in the cytoplasm dependent on histological subtype (**a**) and tumor grading (**b**) of the tumor are shown. (**a**) In non-NST tumors, Gal-7 expression in the cytoplasm is significantly lower than in NST tumors. (**b**) Tumors with G2/3 grading show a significantly higher Gal-7 expression in the cytoplasm compared to G1 tumors. Red asterisks indicate means. Please note that individual datapoints have been jittered to avoid overlap.

**Figure 2 cancers-12-00953-f002:**
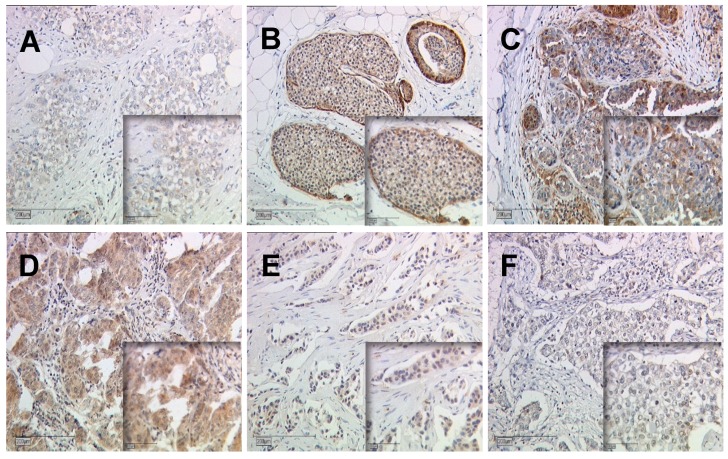
Gal-7 and Gal-8 expression dependent on tumor grading. Exemplary immunohistochemical staining of Gal-7 in grade 1 (**A**), 2 (**B**), and 3 (**C**) tumors and of Gal-8 in grade 1 (**D**), 2 (**E**), 3 (**F**) tumors are shown. Magnification: main images x10, image sections x25.

**Figure 3 cancers-12-00953-f003:**
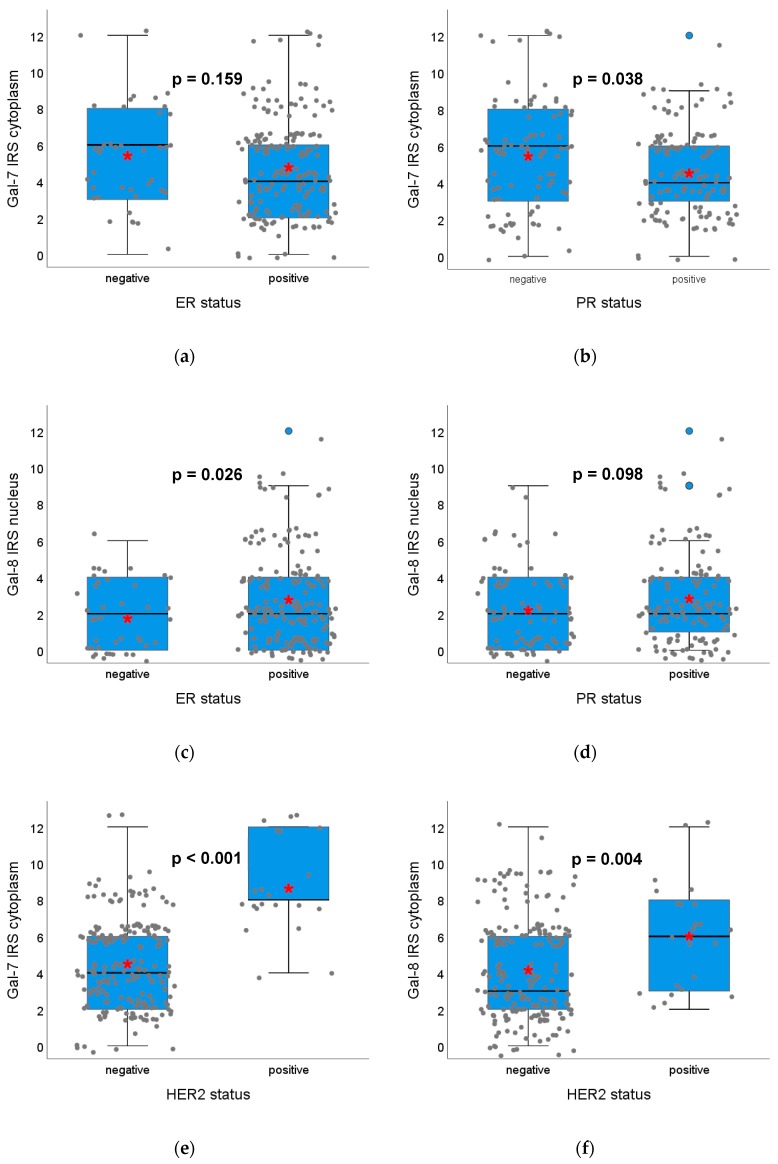
Association of Gal-7 and Gal-8 expression with the ER-, PR- and HER2-status. Boxplots of the median IRS of Gal-7 staining in the cytoplasm dependent on ER-status (**a**), PR-status (**b**) and HER2-status (**e**) and of Gal-8 staining in the nucleus dependent on ER-status (**c**) and PR-status (**d**) as well as Gal-8 staining in the cytoplasm dependent on HER2-status (**f**) are shown. ER-positive tumors do not differ concerning Gal-7 staining but show a higher Gal-8 staining. PR-positive tumors show lower Gal-7 staining and a trend towards higher Gal-8 staining. HER2-positive tumors show a significantly higher Gal-7 and Gal 8 expression in the cytoplasm compared to HER2-negative tumors. Staining in the nucleus does not show significant differences. Red asterisks indicate means. Please note that individual data points have been jittered to avoid overlap.

**Figure 4 cancers-12-00953-f004:**
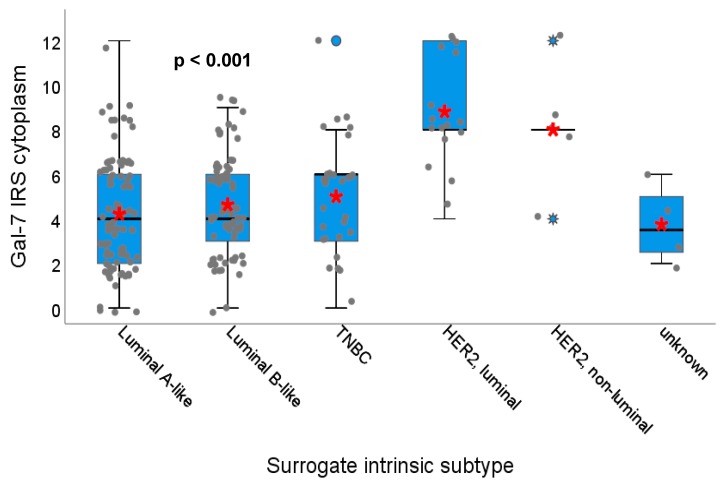
Association of Gal-7 expression with different surrogate intrinsic subtype. Boxplot of the median IRS of Gal-7 staining in the cytoplasm dependent on the surrogate intrinsic subtype of the tumor is shown. HER2-positive, both luminal and non-luminal tumors show a significantly higher Gal-7 expression in the cytoplasm compared to the other subtypes. Red asterisks indicate means. Please note that individual datapoints have been jittered to avoid overlap.

**Figure 5 cancers-12-00953-f005:**
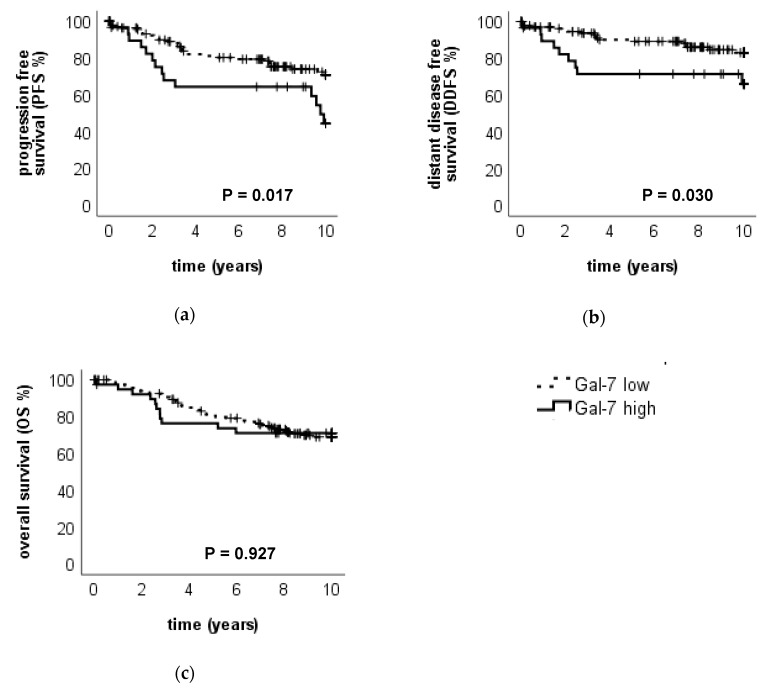
Association of Gal-7 expression in the cytoplasm to the clinical outcome. Kaplan–Meier analysis of PFS (**a**), DDFS (**b**) and OS (**c**) in Gal-7 high- and low-expressing tumors (in the cytoplasm) is shown. Tumors with high Gal-7 expression in the cytoplasm showed a significantly impaired PFS and DDFS.

**Figure 6 cancers-12-00953-f006:**
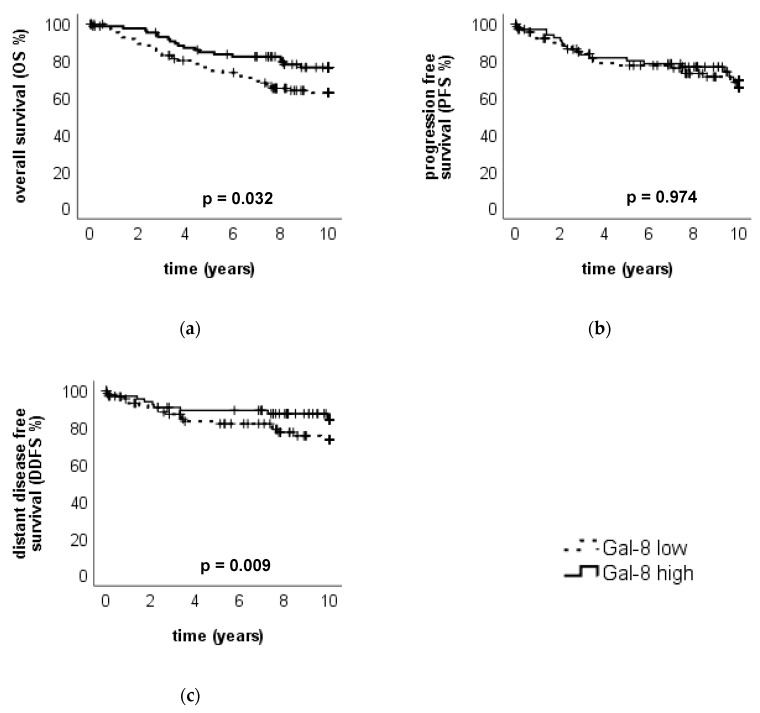
Association of Gal-8 expression in the cytoplasm to the clinical outcome. Kaplan–Meier analysis of OS (**a**), PFS (**b**) and DDFS (**c**) in Gal-8 high- and low-expressing tumors (in the cytoplasm) is shown. Tumors with high Gal-8 expression in the cytoplasm showed a significantly improved OS.

**Figure 7 cancers-12-00953-f007:**
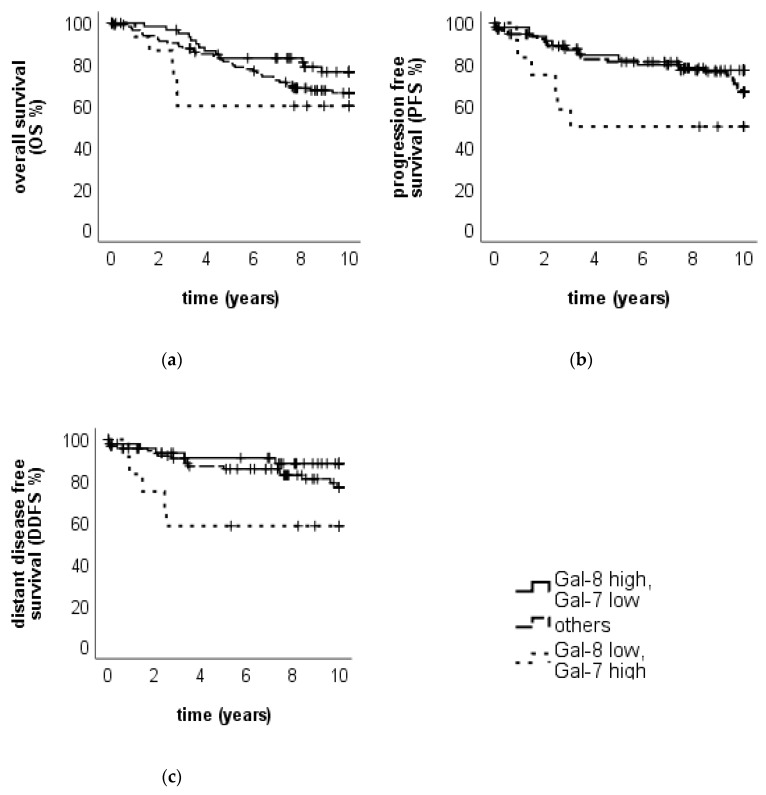
OS, PFS and DDFS for the two groups. Kaplan–Meier analysis of OS (**a**), PFS (**b**) and DDFS (**c**) in tumors with combined high or low Gal-7 expression with low or high Gal-8 expression (in the cytoplasm), respectively, are shown. Tumors with high Gal-7 and low Gal-8 expression in the cytoplasm show the trend of an impaired OS, while tumors with high Gal-8 and low Gal 7 expression show the trend of an improved OS. Tumors with high Gal-7 and low Gal-8 expression in the cytoplasm show the trend of an impaired PFS and a significantly reduced DDFS, while tumors with high Gal-8 and low Gal-7 expression show the trend of an improved PFS and a DDFS.

**Figure 8 cancers-12-00953-f008:**
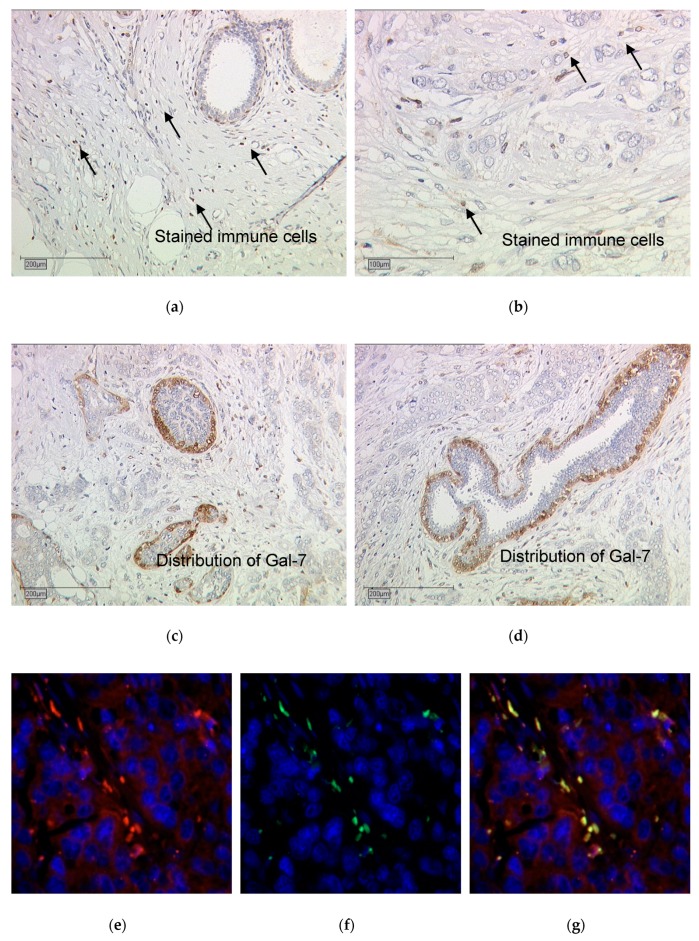
Gal-7-staining results (**a**–**d**) and double immunofluorescence staining with Gal 7 and CD68 (**e**–**g**). Exemplary pictures of Gal 7-staining results are shown, revealing the distribution of Gal 7-staining in tumor cells (**a**,**b**). Stained immune cells in Gal 7-expressing tumors are shown in (**c**) and (**d**). Immunofluorescence results: Gal-7 is shown in red (**e**), CD68 in green (**f**) and the nucleus is stained with DAPI in blue. Picture (**g**) shows that Gal 7- and CD8-staining is overlapping in some cells, which appear yellow, showing that these are macrophages. Magnification: Images 8a, 8c and 8d x10, and images 8b, 8e, 8f and 8g x25.

**Table 1 cancers-12-00953-t001:** Staining results of Gal-7 in the cytoplasm and the nucleus. Gal = galectin; NA = not applicable; IRS = immunoreactivity score.

Gal-7 Cytoplasm			Gal-7 Nucleus		
IRS	n	%	IRS	n	%
0	7	3.0	0	35	14.9
1	2	0.9	1	17	7.2
2	43	18.3	2	64	27.2
3	21	8.9	3	51	21.7
4	40	17.0	4	25	10.6
6	60	25.5	6	24	10.2
8	27	11.5	NA	19	8.1
9	8	3.4	total	235	100.0
12	8	3.4			
NA	19	8.1			
total	235	100.0			

**Table 2 cancers-12-00953-t002:** Staining results of Gal-8 in the cytoplasm and the nucleus. Gal = galectin; NA = not applicable; IRS = immunoreactivity score.

Gal-8 Cytoplasm			Gal-8 Nucleus		
IRS	n	%	IRS	n	%
0	15	6.4	0	63	26.8
1	12	5.1	1	14	6.0
2	50	21.3	2	55	23.4
3	27	11.5	3	7	3.0
4	18	7.7	4	43	18.3
6	54	23.0	6	22	9.4
8	16	6.8	8	1	0.4
9	18	7.7	9	9	3.8
12	5	2.1	12	1	0.4
NA	20	8.5	NA	20	8.5
total	235	100.0	total	235	100.0

**Table 3 cancers-12-00953-t003:** Spearman correlation analysis of Gal-7 and Gal-8 expression in the nucleus and cytoplasm. Significant correlations are displayed in bold. ** indicates a significance level of *p* < 0.01.

		Gal-7 IRS Cytoplasm	Gal-7 IRS Nucleus	Gal-8 IRS Cytoplasm	Gal-8 IRS Nucleus
**Gal-7 IRS cytoplasm**	Correlation Coefficient	1.000	**0.467 ****	**0.332 ****	**0.188 ****
	Sig. (2-tailed)		<0.001	<0.001	0.007
	N	216	216	206	206
**Gal-7 IRS nucleus**	Correlation Coefficient	**0.467 ****	1.000	**0.185 ****	**0.301 ****
	Sig. (2-tailed)	<0.001		0.008	<0.001
	N	216	216	206	206
**Gal-8 IRS cytoplasm**	Correlation Coefficient	**0.332 ****	**0.185 ****	1.000	**0.594 ****
	Sig. (2-tailed)	<0.001	0.008		0.000
	N	206	206	215	215
**Gal-8 IRS nucleus**	Correlation Coefficient	**0.188 ****	**0.301 ****	**0.594 ****	1.000
	Sig. (2-tailed)	0.007	<0.001	<0.001	
	N	206	206	215	215

**Table 4 cancers-12-00953-t004:** Multivariate analysis of PFS concerning Gal 7 expression in the cytoplasm. Significant factors are highlighted in bold.

Prognostic Factor	B	SE	Wald	df	Sig.	Exp(B)	95,0% CI for Exp(B)	
							Lower	Upper
Histological subtype (NST vs. non-NST)	0.245	0.459	0.286	1	0.593	1.278	0.520	3.140
Grading (G1 vs. G2-3)	0.312	0.817	0.146	1	0.702	1.367	0.276	6.773
Tumor size (pT1 vs. pT2-4)	0.284	0.377	0.564	1	0.453	1.328	0.634	2.783
Nodal status (pN0 vs. pN1-3)	0.553	0.397	1.935	1	0.164	1.738	0.798	3.788
HER2 status (positive vs. negative)	−0.161	0.538	0.090	1	0.764	0.851	0.297	2.443
ER (positive vs. negative)	−0.823	0.459	3.219	1	0.073	0.439	0.179	1.079
PR (positive vs. negative)	−0.345	0.460	0.565	1	0.452	0.708	0.288	1.743
Patient age (continuous)	0.010	0.017	0.329	1	0.566	1.010	0.977	1.043
**Gal-7 expression in the cytoplasm (high vs. low)**	0.806	0.410	3.860	1	**0.049**	2.239	1.002	5.003

**Table 5 cancers-12-00953-t005:** Cox regression analysis of prognostic factors for OS in breast cancer patients. Significant factors are highlighted in bold.

Prognostic Factor	B	SE	Wald	df	Sig.	Exp(B)	95,0% CI for Exp(B)	
							Lower	Upper
Histological subtype (NST vs. non-NST)	0.199	0.456	0.190	1	0.663	1.220	0.499	2.983
Grading (G1 vs. G2-3)	−0.321	0.591	0.295	1	0.587	0.725	0.228	2.311
Tumor size (pT1 vs. pT2-4)	0.237	0.385	0.378	1	0.539	1.267	0.596	2.693
**Nodal status (pN0 vs. pN1-3)**	1.144	0.406	7.927	1	**0.005**	3.140	1.416	6.964
HER2 status (positive vs. negative)	0.429	0.617	0.483	1	0.487	1.535	0.458	5.143
**ER (positive vs. negative)**	−1.178	0.504	5.466	1	**0.019**	0.308	0.115	0.827
PR (positive vs. negative)	0.568	0.483	1.381	1	0.240	1.765	0.684	4.551
**Patient age (continuous)**	0.051	0.017	9.367	1	**0.002**	1.053	1.019	1.088
Gal-8 expression in the cytoplasm (high vs. low)	−0.258	0.372	0.479	1	0.489	0.773	0.372	1.603

**Table 6 cancers-12-00953-t006:** Patients’ characteristics.

Patients‘ Characteristics	Median	SD
**Age**	58.2	13.3
	**N**	**%**
**Histological subtype**		
NST	126	53.6
Non-NST	96	40.9
NA	13	5.5
**Intrinsic surrogate subtype**		
Luminal A-like	103	43.8
Luminal B-like	73	31.1
HER2-positive, luminal	17	7.2
HER2-positive, non-luminal	7	3.0
TNBC	31	13.2
NA	4	1.7
**Grading**		
Grade 1	17	7.2
Grade 2	90	38.3
Grade 3	55	23.4
NA	73	31.1
**Lymph node involvement (pN)**		
pN0	128	54.5
pN1	87	37.0
pN2	10	4.3
NA	10	4.3
**Tumor size (pT)**		
pT1 (≤2 cm)	160	68.1
pT2 (2–5 cm)	68	28.9
pT3 (>5 cm)	1	0.4
pT4 (with infiltration in the epidermis or the thoracic wall)	5	2.1
NA	1	0.4
**HER2 status**		
Positive	24	10.2
Negative	208	88.5
NA	3	1.3
**ER status**		
Positive	192	81.7
Negative	43	18.3
**PR status**		
Positive	141	60.0
Negative	94	40.0

NST = no special type, NA = not available, ER = estrogen receptor, PR = progesterone receptor, HER2 = human epidermal growth factor receptor 2, TNBC = triple negative breast cancer.

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
