# Peer review of "High Galectin-7 and Low Galectin-8 Expression and the Combination of both are Negative Prognosticators for Breast Cancer Patients"

_cancers, 2020, doi:10.3390/cancers12040953_

Round 1
Reviewer 1 Report
The manuscript is well written. In figures 1, 3 and 4, the standard deviation is very high to consider the statistical significance. Hence, consider re-graphing using individual value plots.
Author Response
Response letter to Reviewer 1
The manuscript is well written. In figures 1, 3 and 4, the standard deviation is very high to consider the statistical significance. Hence, consider re-graphing using individual value plots.
We thank you for this comment, the standard deviation is indeed very high. We tried to re-graph our results and changed the figures 1, 3 and 4 in the manuscript as you suggested. We hope our data are now presented in a more appropriate way.

Reviewer 2 Report
This is an immunohistopathological study of the expression of galectin-7 and -8 in samples of breast cancer. The expression of each galectin in cytosol or nucleus is scored in a semiquantitative way, and different types of breast cancer compared. The most striking finding is that galectin-7 expression in cytosol appears elevated in HER2 positive cancers compared to others. Otherwise differences are fairly subtle, especially considering the rough semiquantitative scoring involving a combination of staining intensity and fraction of positive cells. The survival curves of patient groups with different scores for the two galectins are also compared. Here the most striking fining is that galectin-7 correlates with a poor prognosis in several end points, and that the combination of high galectyin-7 and low galectin-8 gives an even poorer prognosis in a small subgroup of patients. Technically the study appears carefully and well done, so worth reporting. However, some details needs clarification. Moreover, it is all phenomelogical so Discussion could be shortened and made a bit more insightful.
Line 390 – 394. It is unclear if and when a secondary antibody was included in the staing. Is it part of the post block reagent? What titer? What species?
Line 427 – 443. Statistical analysis. For the Kaplan-Meyer plots, it should be briefly described what were the censored cases, as they affect the analysis of significance.
The description of the use of ROC plots to define cutoff is also hard to follow. An example should be included in the supplementary material.
Line 266 – 283. The immune cells. Were galectin expression in these included as part of the scoring for the cancer specimen, or excluded?
The Discussion is not very insightful, so could be shortened. It contains a number of references to previous work, but with some exception not very clear connection to present data. Nevertheless, here are three suggested additional references that might be of interest, and the author can include if they want to.
1) Bhat et al. Nuclear repartitioning of galectin-1 by an extracellular glycan switch regulates mammary morphogenesis. Proc Natl Acad Sci U S A. 2016 Aug 16;113(33):E4820-7. doi: 10.1073/pnas.1609135113. Epub 2016 Aug 5.
This interesting paper describes a mechanism and function on how the nuclear-cytosolic distribution of galectin-1 is regulated in the mammary gland.
2) Huflejt and Leffler. Galectin-4 in normal tissues and cancer. Glycoconjugate J. 2003;20(4):247-55.
This paper describes a comparison of galectin expression in cancer cell lines, showing that low differentiation correlates with high expression of galectin-1, and low of other galectins, whereas high differentiation correlates with high expression of galectin-4 and low of galectin-1. As galectin-4 belongs to same group as galectin-8 and galectin-1 same as -7, this might be relevant for present study. Could one see a difference within a tumor tissue between more highly differentiated cancer cells, and less differentiated ones?
In this paper is also some immunohistopathology of galectin-4 expression in breast cancer.
3) Carlsson et al. Different fractions of human serum glycoproteins bind galectin-1 or galectin-8, and their ratio may provide a refined biomarker for pathophysiological conditions in cancer and inflammatory disease. BBA-Gen Subjects. 2012;1820(9):1366-72.
This paper describes that glycoprotein ligands for galectin-1 in serum are strongly elevated in breast cancer patients, whereas glycoprotein ligands for galectin-8 are decreased, making the ratio of the two a strong marker for breast cancer vs, patients with inflammatory disease or healthy controls. Even if this is about galectin ligands, and not the galectins themselves, it might be an interesting parallel to the present findings.
Author Response
Response letter to Reviewer 2
Line 266 – 283. The immune cells. Were galectin expression in these included as part of the scoring for the cancer specimen, or excluded?
We thank you for this comment that it was not absolutely clear what was included to score the expression in the cancer specimen. To determine galectin expression in the cancer specimen, we counted only the percentage of positively stained tumor cells, not other cells like immune cells. The observation, that also immune cells stained positive for galectin-7 was an additional observation which led to the further analysis described in paragraph 3.4. To make this clearer to the reader, we added a respective sentence in lines 271-272 (results) and lines 404-407 (methods).
Line 390 – 394. It is unclear if and when a secondary antibody was included in the staing. Is it part of the post block reagent? What titer? What species?
The secondary antibodies are indeed part of the Reagent 3, the HRP polymer. The species is mouse and rabbit. As the antibodies are part of the provided reagent, the exact titers are not provided by the manufacturer. This information has been added to the manuscript (lines 391-395).
Line 427 – 443. Statistical analysis. For the Kaplan-Meyer plots, it should be briefly described what were the censored cases, as they affect the analysis of significance.
The description of the use of ROC plots to define cutoff is also hard to follow. An example should be included in the supplementary material
In the Kaplan Meier plots, censored cases are cases for which the second event is not recorded (for example, people still alive at the end of the study). The Kaplan-Meier procedure is a method of estimating time-to-event models in the presence of censored cases. The Kaplan-Meier model is based on estimating conditional probabilities at each time point when an event occurs and taking the product limit of those probabilities to estimate the survival rate at each point in time. This information has been added to the manuscript (lines 437-438).
Thank you for remarking that it is hard to follow the use of ROC plot. We included an example in the supplementary material (Table S5) as you suggested and added a complete description on defining the cutoff in lines 448-449.
The Discussion is not very insightful, so could be shortened. It contains a number of references to previous work, but with some exception not very clear connection to present data. Nevertheless, here are three suggested additional references that might be of interest, and the author can include if they want to.
Thank you for your suggestions, we tried to shorten the discussion section and added your recommended references. We aimed to focus on the main points of the study that you mentioned in your comment: the key points that Gal-7 expression is associated to HER2 positive breast cancer, that elevated Gal-7 expression is a negative prognosticator and that the combination of both Gal-7 and Gal-8 might serve as a prognostic marker tool. We left the additional marginal results out of the discussion. However, we still put some focus on possible mechanisms of galectin regulation and the role of intra- and extracellular distribution and nuclear accumulation, as we felt discussing underlying mechanisms is essential to understand the relevance of a prognostic factor. We also discussed therapeutic options more in detail, as this was additionally suggested by reviewer 3. We hope that the connection to present data is now comprehensible.
1) Bhat et al. Nuclear repartitioning of galectin-1 by an extracellular glycan switch regulates mammary morphogenesis. Proc Natl Acad Sci U S A. 2016 Aug 16;113(33):E4820-7. doi: 10.1073/pnas.1609135113. Epub 2016 Aug 5.
This interesting paper describes a mechanism and function on how the nuclear-cytosolic distribution of galectin-1 is regulated in the mammary gland.
This is an interesting study, we inserted this information in lines 327-331. We used this study to discuss underlying mechanisms of galectin regulation.
2) Huflejt and Leffler. Galectin-4 in normal tissues and cancer. Glycoconjugate J. 2003;20(4):247-55.
This paper describes a comparison of galectin expression in cancer cell lines, showing that low differentiation correlates with high expression of galectin-1, and low of other galectins, whereas high differentiation correlates with high expression of galectin-4 and low of galectin-1. As galectin-4 belongs to same group as galectin-8 and galectin-1 same as -7, this might be relevant for present study. Could one see a difference within a tumor tissue between more highly differentiated cancer cells, and less differentiated ones?
In this paper is also some immunohistopathology of galectin-4 expression in breast cancer.
We discussed this paper and your reflections in the lines 320-323. To discuss this paper, we added our result that high galectin-1 expression is associated with higher grading (and therefore lower cell differentiation) to the discussions section.
3) Carlsson et al. Different fractions of human serum glycoproteins bind galectin-1 or galectin-8, and their ratio may provide a refined biomarker for pathophysiological conditions in cancer and inflammatory disease. BBA-Gen Subjects. 2012;1820(9):1366-72.
This paper describes that glycoprotein ligands for galectin-1 in serum are strongly elevated in breast cancer patients, whereas glycoprotein ligands for galectin-8 are decreased, making the ratio of the two a strong marker for breast cancer vs, patients with inflammatory disease or healthy controls. Even if this is about galectin ligands, and not the galectins themselves, it might be an interesting parallel to the present findings.
Thank you for this important and useful paper, we inserted it in the lines 314-316.

Reviewer 3 Report
Authors show clinical outcome patients showing both, high Gal-7 combined with low Gal-8 expression, was very poor. These are few comments authors can address.
- As authors mentioned many galectins are involved in oncogenesis, are these specific to gal-7 and 8?
- Every year 300,000 breast cancer patients are diagnosed world wide, authors performed this study in a cohort of 235 patients, does these small cohort have can represent the huge spectrum of breast cancer?
- Can authors comment if Gal-7 and 8 will effect the treatment status of the patients or if this are only prognostic factors, because galectins are shown to be influence therapeutic potential in some studies
Author Response
Response letter to Reviewer 3
As authors mentioned many galectins are involved in oncogenesis, are these specific to gal-7 and 8?
Thank you for your further interest in this topic. Many galectines are involved in oncogenesis, as described in the introduction and in the results section. We have already mentioned some known evidence about galectin-3 and galectin-1 and added further information (lines 64-70) in the introduction and also discussed the poor knowledge about the role of Gal‑8 in tumorogenesis (lines 81-84). The specific role of Gal‑7 in breast cancer was also described in the introduction (lines 70-79) and we added the information that it is not known if these effects are specific to Gal-7, as most of the studies concentrated on one galectin and did not compare all galectins among each other (lines 79-81).
Every year 300,000 breast cancer patients are diagnosed world wide, authors performed this study in a cohort of 235 patients, does these small cohort have can represent the huge spectrum of breast cancer?
Our cohort represents a classical clinical cohort of BC patients with a long follow up period of up to 10 years. In terms of the distribution of clinical and pathological characteristics and tumor biology it reflects the spectrum of breast cancer. We also added this information to the manuscript (lines 359-360). We can analyze this cohort after a long follow up period, however, long follow up periods come at the cost that due to the rapid changes in oncology this cohort is not treated with today’s standard of therapy. As therapy in oncology changes so quickly, it is simply not possible to have both, long follow up periods and a cohort treated with therapies that are still standard of care at the time point of the analysis. Therefore, we suggested to study these findings on a more recent cohort as mentioned in the conclusion (lines 467-468).
Can authors comment if Gal-7 and 8 will effect the treatment status of the patients or if this are only prognostic factors, because galectins are shown to be influence therapeutic potential in some studies
We already had the information in the manuscript that first attempts in targeting Gal-7 have already been made. To our knowledge, there has no study been so far that targets Gal-8. We added some information about the therapeutic potential that has been shown for Gal-3 and Gal‑1 and speculated about a therapeutic potential of Gal-7, as it belongs to the same family as Gal-1 (lines 346-353).
As you already suggested improvement of English language, we performed additional proofreading of the manuscript.
